# A national household survey on HIV prevalence and clinical cascade among children aged ≤15 years in Kenya (2018)

**Immaculate Mutisya**[1]\*, **Evelyn Muthoni**[2], **Raphael O. Ondondo**[1], **Jacques Muthusi**[1], **Lennah Omoto**[1], **Charlotte Pahe**[2], **Abraham Katana**[1], **Evelyn Ngugi**[1], **Kenneth Masamaro**[1], **Leonard Kingwara**[2], **Trudy Dobbs**[3], **Megan Bronson**[3], **Hetal K. Patel**[3], **Nicholas Sewe**[2], **Doris Naitore**[4], **Kevin De Cock**[1], **Catherine Ngugi**[2], **Lucy Nganga**[1]

1 Division of Global HIV & TB, Center for Global Health, U.S. Centers for Disease Control and Prevention, Nairobi, Kenya, 2 National AIDS & STI Control Programme, Ministry of Health, Nairobi, Kenya, 3 Division of Global HIV & TB, Center for Global Health, U.S. Centers for Disease Control and Prevention, Atlanta, GA, United States of America, 4 International Center for AIDS Care and Treatment Programs, Nairobi, Kenya

\* yry1@cdc.gov

## Abstract

We analyzed data from the 2018 Kenya Population-Based HIV Impact Assessment (KENPHIA), a cross-sectional, nationally representative survey, to estimate the burden and prevalence of pediatric HIV infection, identify associated factors, and describe the clinical cascade among children aged < 15 years in Kenya. Interviewers collected information from caregivers or guardians on child's demographics, HIV testing, and treatment history. Blood specimens were collected for HIV serology and if HIV-positive, the samples were tested for viral load and antiretrovirals (ARV). For participants <18 months TNA PCR is performed. We computed weighted proportions with 95% confidence intervals (CI), accounting for the complex survey design. We used bivariable and multivariable logistic regression to assess factors associated with HIV prevalence. Separate survey weights were developed for interview responses and for biomarker testing to account for the survey design and non-response. HIV burden was estimated by multiplying HIV prevalence by the national population projection by age for 2018. Of 9072 survey participants (< 15 years), 87% (7865) had blood drawn with valid HIV test results. KENPHIA identified 57 HIV-positive children, translating to an HIV prevalence of 0.7%, (95% CI: 0.4%–1.0%) and an estimated 138,900 (95% CI: 84,000–193,800) of HIV among children in Kenya. Specifically, children who were orphaned had about 2 times higher odds of HIV-infection compared to those not orphaned, adjusted Odds Ratio (aOR) 2.2 (95% CI:1.0–4.8). Additionally, children whose caregivers had no knowledge of their HIV status also had 2 times higher odds of HIV-infection compared to whose caregivers had knowledge of their HIV status, aOR 2.4 (95% CI: 1.1–5.4)". From the unconditional analysis; population level estimates, 78.9% of HIV-positive children had known HIV status (95% CI: 67.1%–90.2%), 73.6% (95% CI: 60.9%–86.2%) were receiving ART, and 49% (95% CI: 32.1%–66.7%) were virally suppressed. However, in the clinical cascade for HIV infected children, 92% (95% CI: 84.4%–100%) were receiving ART, and of these, 67.1% (95% CI: 45.1%–89.2%) were virally suppressed. The KENPHIA

---

requested from: https://phia-data.icap.columbia.edu/datasets?country_id=14.

**Funding:** This survey was supported by the U.S. President's Emergency Plan for AIDS Relief (PEPFAR) through the United States Centers for Disease Control and Prevention (CDC) under the terms of Cooperative Agreement #U2GGH001226. Additional support was provided by United States Agency for International Development (USAID) and the Global Fund to Fight AIDS, TB and Malaria." There was no additional external funding received for this study.

**Competing interests:** The authors have declared that no competing interests exist.

survey confirms a substantial HIV burden among children in Kenya, especially among orphans.

## Introduction

An estimated 1.7 million children aged <15 years were living with HIV globally at the end of 2018 [1]. Despite widespread availability of prevention of mother-to-child transmission (PMTCT) services and life-saving antiretroviral treatment (ART) for children living with HIV (CLHIV), there were 160,000 new HIV infections and 100,000 AIDS-related deaths among children aged <15 years globally [1, 2]. Nine of ten CLHIV aged 0–14 years live in sub-Saharan Africa [1], where persistent gaps in PMTCT, early infant diagnosis, and pediatric ART coverage have continued to undermine access to treatment and efforts to reduce AIDS-related deaths among CLHIV [1, 3]. In 2018, maternal ART coverage in sub-Saharan Africa was 92%, and the mother-to-child transmission (MTCT) rate was 9%. Despite high MTCT, only 68% of HIV-exposed infants in the region received an HIV test within 2 months of birth as recommended by the World Health Organization (WHO) [4]. Furthermore, only 54% of CLHIV received ART [5].

In Kenya, an estimated 1.51 million people, including 105,000 (range, 95,000–160,000) children were living with HIV by the end of 2018 [6]. Although pediatric HIV infections have decreased substantially in Kenya in recent years, the MTCT rate has remained high at 10.8%, with approximately 6800 new infections annually—predominantly among infants born to women who had never received ART or who had interruptions in ART during pregnancy [6]. In Kenya, an estimated 690,000 children are orphaned by AIDS [5], and orphaned CLHIV live under relatives as caregivers or children homes.

In 2015, Kenya adopted the Joint United Nations Programme on HIV/AIDS (UNAIDS) 90-90-90 targets for epidemic control by 2020, aiming to diagnose 90% of all HIV-positive people, provide sustained ART to 90% of those with a known HIV diagnosis, and to have an undetectable viral load (VL) among 90% of those receiving ART. Since then, Kenya has rapidly scaled up adult and pediatric HIV care and treatment services [6]. A nationally representative HIV indicator survey conducted in 2012 –the Kenya AIDS indicator survey 2012 (KAIS 2012), showed an estimated HIV prevalence of 0.9% among children aged 18 months –14 years and identified key programmatic gaps [7].

The 2018 Kenya Population-based HIV Impact Assessment (KENPHIA) evaluated the national HIV program, identified programmatic gaps, and monitored epidemic control in Kenya. We analyzed data from the 2018 KENPHIA participants aged 0–14 years to estimate HIV prevalence and the total number of CLHIV nationally; to assess knowledge of HIV status, ART coverage, and VL suppression among CLHIV; and to identify factors associated with HIV infection.

## Materials and methods

### Survey design

KENPHIA 2018 was a cross-sectional household survey that used a two-stage cluster sample of 800 clusters selected from 5360 clusters contained in the Kenya National Bureau of Statistics (KNBS), Fifth National Sample Survey and Evaluation Programme (NASSEP V) sample [8]. The survey targeted 18,362 households; in every third household, all children aged 0–14 years were recruited into the survey. A household sociodemographic questionnaire was

administered to the household heads after they had provided written informed consent. Parent/guardian and emancipated minors provided written consent to participate in the survey. Additionally, all children aged ≥12 years provided written assent to participate. To mitigate potential information bias and participant misclassification, study questionnaires were carefully designed using plain language, and conducted in the participant's preferred language by trained interviewers.

## Laboratory methods

All children aged 0–14 years were eligible for survey participation survey. Survey staff collected a venous blood sample for all children aged ≥2 years; for children aged <2 years, survey staff collected capillary whole-blood samples by finger prick or heel prick (< 6 months) into microtainer EDTA tubes [9]. Blood samples from children aged ≥18 months were tested for HIV infection at the household, per Kenya National HIV Testing Services Guidelines, using Determine HIV 1/2 (Abbott Molecular Inc.) and First Response HIV 1-2.O (Premier Medical Corporation) HIV rapid test kits and results were provided onsite (S1A Fig). For children aged <18 months, a rapid HIV Determine test was performed to determine HIV exposure (S1B Fig). All HIV exposed infants and children with HIV-positive caregivers or whose caregiver's HIV status was unknown, underwent additional nucleic acid (DNA) testing using GeneXpert point of care (POCT) Qual assay (Cepheid) at a satellite survey laboratory and had dried blood spots processed for DNA PCR confirmatory testing using Cobas AmpliPrep/Cobas TaqMan (CAP/CTM) HIV-1 Qualitative Test, version 2.0 assay (Roche Molecular Diagnostics) at the central survey laboratory (National HIV Reference Laboratory) as shown in S1B Fig.

Samples were further processed per protocol as DBS and/or plasma specimens and stored until further testing is conducted. HIV VL and antiretroviral (ARV) testing was performed for all HIV-positive participants. All results from satellite and central survey laboratories were returned to the child's parent/guardian at the household within 7 days. All HIV-exposed children who were HIV-negative or confirmed HIV-positive were referred to HIV clinics of their choice for care or follow-up. To determine HIV prevalence in the survey, we subjected all HIV-positive specimens by serology (i.e., for children aged ≥18 months) to a confirmatory test using Genius HIV-1/2 supplemental assay (Bio-Rad Laboratories). For all HIV-positive participants, HIV VL testing was performed using automated Roche CAP/CTM HIV-1 RNA quantitation test (Roche Molecular Diagnostics); VL suppression was defined as <1000 copies per mL of blood in the survey specimen.

## Variable definitions

A caregiver was defined as either a biological or non-biological parent or guardian who was taking care of the child during KENPHIA. A child was considered an orphan if both the mother and father were dead. Knowledge of HIV status was defined as "known" if the caregiver had confirmed that the child was HIV positive or if antiretroviral metabolites were detected in the child's blood. HIV status was defined as "unknown" if caregivers were unaware of their own or the child's HIV status. An HIV-exposed infant was defined as a child born of or under the care of an HIV-positive caregiver. Infants aged ≤18 months were confirmed to be HIV positive if the HIV PCR test result was positive, whereas children aged >18 months were confirmed to be HIV positive if they had a positive HIV antibody test result. A child was defined as receiving antiretroviral drugs if this information was provided by the caregiver or if these drugs were detected in the child's blood.

### Data analysis

We computed frequencies and weighted proportions for factors and medians for continuous variables. These were compared across age categories. HIV prevalence was calculated as the proportion of children aged <15 years who tested positive during the survey. Pearson chi-square tests were used to compare differences across age categories for categorical factors, whereas Kruskal-Wallis tests were used to compare the distribution of medians across the age categories. Factors associated with HIV infection were identified by fitting bivariable logistic regression for HIV prevalence. Associations with p-values <0.05 were considered statistically significant in the multivariable logistic regression analysis. Age and sex, as known confounding variables, were selected *a priori* for inclusion in the multivariable model. Survey weights for all estimates were calculated and adjusted for individual and HIV testing. Analysis was performed using SAS (SAS Institute Inc) statistical software (version 9.4). The number of HIV-positive children aged 0–14 years living in Kenya was calculated by multiplying the prevalence of HIV by the national population projection by age for 2018 [10]. The protocol was reviewed and cleared by the Institutional Review Boards at Kenya Medical Research Institute, Columbia University, and the Centers for Disease Control and Prevention (CDC).

## Results

### Characteristics of study participants

Of 9426 eligible children, 9072 (96.2%) provided written consent and/or assent to participate in the survey. Of the participants, 87% (7865) had blood drawn and valid HIV and VL test results and were included in our analysis (S2 Fig). Half (50.5%) were boys, and median age was 6.1 years (interquartile range [IQR]: 2.6–9.7 years). A larger proportion of participants were aged 0–4 years (36.7%), followed by 5–9 years (33.6%) and 10–14 years (29.5%). More than two-thirds (71.9%) resided in rural areas, and one in ten (10.3%) were orphans (Table 1). Overall, 78% of caregivers had knowledge of HIV status; of these, 5.5% reported an HIV-positive test result (results not shown). Three-quarters (75.1%) of children had never been tested for HIV, and 21.4% had undergone HIV testing before the survey (Table 1).

### HIV prevalence and estimated burden

A total of 57 children tested positive for HIV, translating to an overall HIV prevalence of 0.7% (95% CI: 0.4%–1.0%). Although HIV prevalence among children aged 10–14 years (1.1% [95% CI: 0.5%–1.6%]) was higher than among those aged 0–4 years (0.4% [95% CI: 0.1%–0.6%]), this difference was not statistically significant. HIV prevalence was comparable between boys (0.8% [95% CI: 0.4%–1.1%]) and girls (0.7% [95% CI: 0.2%–1.1%]). HIV prevalence among children who had ever been tested (2.8% [95% CI: 1.6%–4.0%]) than among those who had never been tested (0.1% [95% CI: 0.0%–0.2%]) or those with unknown testing history (1.1% [95% CI: 0.0%–3.4%]; Table 2). Based on the overall HIV prevalence (0.7%), we estimated the number of children aged 0–14 years living with HIV in Kenya at 138,900 (84,000–193,800), with girls accounting for 46% (n = 64,300 [95% CI: 22,900–105,600]) of the total estimated burden.

### Factors associated with HIV infection

Table 3 shows logistic regression results for factors associated with HIV prevalence among children aged 0–14 years. On bivariable analysis, children aged 10–14 years had 3 times higher odds of having an HIV-positive test result (unadjusted odds ratio [OR] 2.8; 95% CI: 1.2–6.8) compared to children aged 0–4 years. Children who were orphans had higher odds to be HIV

**Table 1. Characteristics of children aged 0–14 years by age group-KENPHIA, 2018.**

| Characteristic | 0–4 years | | 5–9 years | | 10–14 years | | Total | |
| --- | --- | --- | --- | --- | --- | --- | --- | --- |
| | Unweighted n (Weighted %) | 95% CI | Unweighted n (Weighted %) | 95% CI | Unweighted n (Weighted %) | 95% CI | Unweighted N (Weighted %) | 95% CI |
| **Total** | **2457 (36.9)** | **36.9–36.9** | **2847 (33.6)** | **33.6–33.6** | **2561 (29.5)** | **29.5–29.5** | **7865 (100)** | |
| **Sex** | | | | | | | | |
| Male | 1320 (50.5) | 50.5–50.5 | 1469 (50.5) | 50.5–50.5 | 1301 (50.6) | 50.6–50.6 | 4090 (50.5) | 50.5–50.5 |
| Female | 1137 (49.5) | 49.5–49.5 | 1378 (49.5) | 49.5–49.5 | 1260 (49.4) | 49.4–49.4 | 3775 (49.5) | 49.5–49.5 |
| **Residence** | | | | | | | | |
| Urban | 734 (30.4) | 27.3–33.6 | 802 (27.3) | 24.0–30.6 | 736 (26.2) | 23.4–29.1 | 2272 (28.1) | 25.6–30.7 |
| Rural | 1723 (69.6) | 66.4–72.7 | 2045 (72.7) | 69.4–76.0 | 1825 (73.8) | 70.9–76.6 | 5593 (71.9) | 69.3–74.4 |
| **Orphaned/vulnerable child** | | | | | | | | |
| Mother, Father, or Both dead | 163 (6.3) | 4.7–7.8 | 302 (10.3) | 8.5–12.2 | 381 (15.4) | 13.0–17.7 | 846 (10.3) | 8.9–11.7 |
| Both parents alive | 2213 (90.3) | 88.7–91.9 | 2474 (86.5) | 84.4–88.5 | 2115 (81.7) | 79.3–84.2 | 6802 (86.5) | 85.1–87.9 |
| Missing | 81 (3.4) | 2.5–4.3 | 71 (3.2) | 2.2–4.3 | 65 (2.9) | 1.8–4.0 | 217 (3.2) | 2.5–3.9 |
| **Child ever tested for HIV** | | | | | | | | |
| Yes | 418 (18.7) | 16.1–21.3 | 556 (22.1) | 19.7–24.6 | 548 (24) | 21.5–26.5 | 1522 (21.4) | 19.6–23.1 |
| No | 1968 (78.8) | 76.1–81.6 | 2142 (74.9) | 72.3–77.5 | 1715 (70.6) | 68.0–73.1 | 5825 (75.1) | 73.3–77.0 |
| Don't know | 45 (2.5) | 1.5–3.4 | 69 (3) | 1.8–4.2 | 112 (5.4) | 4.1–6.7 | 226 (3.5) | 2.7–4.3 |
| Missing | 26 (0.9) | 0.4–1.4 | 80 (2.4) | 1.6–3.1 | 186 (5.7) | 4.3–7.2 | 292 (2.8) | 2.1–3.5 |
| **Caregiver age category (years)** | | | | | | | | |
| 15–24 | 572 (21.7) | 19.3–24.2 | 166 (5) | 3.8–6.1 | 26 (0.8) | 0.3–1.3 | 764 (9.9) | 8.8–11.0 |
| 25–34 | 1158 (47.6) | 44.5–50.7 | 1220 (43.3) | 40.5–46.0 | 725 (28.1) | 25.3–30.9 | 3103 (40.4) | 38.1–42.7 |
| 35–44 | 387 (15.7) | 13.5–18.0 | 670 (24.1) | 21.6–26.7 | 795 (31.8) | 28.9–34.7 | 1852 (23.3) | 21.3–25.3 |
| 45+ | 67 (2.7) | 1.9–3.6 | 229 (6.8) | 5.3–8.3 | 375 (13.4) | 11.4–15.3 | 671 (7.2) | 6.2–8.3 |
| Missing | 273 (12.2) | 10.3–14.1 | 562 (20.8) | 18.5–23.1 | 640 (25.9) | 23.3–28.5 | 1475 (19.1) | 17.5–20.7 |
| **Caregiver HIV status** | | | | | | | | |
| Known | 2139 (86) | 83.7–88.2 | 2222 (76.6) | 74.0–79.2 | 1830 (69.8) | 67.1–72.5 | 6191 (78) | 76.1–80.0 |
| Don't know | 318 (14) | 11.8–16.3 | 625 (23.4) | 20.8–26.0 | 731 (30.2) | 27.5–32.9 | 1674 (22) | 20.0–23.9 |

positive (OR, 5.0; 95% CI: 2.5–10.3) than those whose parents were both living. Children whose caregivers did not know their HIV status had roughly 2-times higher odds of HIV-infection compared to whose parent knew their HIV status, aOR 2.4 (95% CI: 1.1–5.4) (Table 3).

**Table 2. HIV prevalence among children aged 0–14 years by age group, KENPHIA 2018.**

| Characteristic | 0–4 years | | 5–9 years | | 10–14 years | | Total | |
|---|---|---|---|---|---|---|---|---|
| | Unweighted n/N | Weighted Prevalence % (95% CI) | Unweighted n/N | Weighted Prevalence % (95% CI) | Unweighted n/N | Weighted Prevalence % (95% CI) | Unweighted n/N | Weighted Prevalence % (95% CI) |
| **Total** | **11/2457** | **0.4 (0.1–0.6)** | **20/2847** | **0.8 (0.2–1.4)** | **26/2561** | **1.1 (0.5–1.6)** | **57/7865** | **0.7 (0.4–1.0)** |
| **Sex** | | | | | | | | |
| Male | 9/1320 | 0.6 (0.1–1.0) | 8/1469 | 0.5 (0.1–0.8) | 15/1301 | 1.3 (0.5–2.2) | 32/4090 | 0.8 (0.4–1.1) |
| Female | 2/1137 | 0.2 (0.0–0.5) | 12/1378 | 1.1 (0.0–2.2) | 11/1260 | 0.8 (0.2–1.3) | 25/3775 | 0.7 (0.2–1.1) |
| **Residence** | | | | | | | | |
| Urban | 1/734 | 0 (0.0–0.1) | 5/802 | 0.4 (0.0–0.7) | 6/736 | 1.1 (0.0–2.5) | 12/2272 | 0.4 (0.1–0.8) |
| Rural | 10/1723 | 0.5 (0.2–0.9) | 15/2045 | 0.9(0.1–1.7) | 20/1825 | 1 (0.5–1.6) | 45/5593 | 0.8 (0.5–1.2) |
| **Orphaned/ vulnerable child** | | | | | | | | |
| Mother, Father, or Both dead | 2/163 | 1.5 (0.0–3.6) | 7/302 | 2.5 (0.4–4.6) | 11/381 | 3.1 (0.4–5.8) | 20/846 | 2.5 (1.1–4.0) |
| Both parents alive | 9/2213 | 0.3 (0.1–0.5) | 13/2474 | 0.6 (0.1–1.1) | 15/2115 | 0.7 (0.3–1.1) | 37/6802 | 0.5 (0.3–0.8) |
| Missing | 0/81 | . (.—.) | 0/71 | . (.—.) | 0/65 | . (.—.) | 0/217 | . (.—.) |
| **Child ever tested for HIV** | | | | | | | | |
| Yes | 10/418 | 1.9 (0.6–3.2) | 15/556 | 2.8 (0.2–5.4) | 21/548 | 3.8(1.6–5.9) | 46/1522 | 2.8 (1.6–4.0) |
| No | 1/1968 | 0 (0.0–0.1) | 3/2142 | 0.1 (0.0–0.3) | 3/1715 | 0.2 (0.0–0.4) | 7/5825 | 0.1(0.0–0.2) |
| Don't know | 0/45 | . (.—.) | 1/69 | 1.9 (0.0–5.8) | 1/112 | 1.2 (0.0–3.8) | 2/226 | 1.1 (0.0–3.4) |
| Missing | 0/26 | . (.—.) | 1/80 | 1.1 (0.0–3.3) | 1/186 | 0.5 (0.0–1.8) | 2/292 | 0.6 (0.0–1.6) |
| **Caregiver age category(years)** | | | | | | | | |
| 15–24 | 0/572 | . (.—.) | 1/166 | 0.4 (0.0–0.9) | 1/26 | 2.2 (0.0–6.7) | 2/764 | 0.1 (0.0–0.3) |
| 25–34 | 6/1158 | 0.4 (0.0–0.8) | 8/1220 | 0.7 (0.0–1.7) | 5/725 | 0.6(0.0–1.3) | 19/3103 | 0.6 (0.2–1.0) |
| 35–44 | 3/387 | 0.7 (0.0–1.4) | 4/670 | 0.9 (0.0–1.9) | 6/795 | 0.6 (0.0–1.2) | 13/1852 | 0.7 (0.2–1.2) |
| 45+ | 0/67 | . (.—.) | 0/229 | . | 3/375 | 0.8 (0.0–1.9) | 3/671 | 0.4 (0.0–1.0) |
| Missing | 2/273 | 0.5 (0.0–1.4) | 7/562 | 1.1 (0.2–2.0) | 11/640 | 2.2 (0.5–3.9) | 20/1475 | 1.4 (0.6–2.2) |
| **Caregiver HIV status** | | | | | | | | |
| Known | 9/2139 | 0.4 (0.1–0.6) | 12/2222 | 0.7 (0.0–1.4) | 14/1830 | 0.6 (0.2–1.0) | 35/6191 | 0.5 (0.2–0.8) |
| Don't know | 2/318 | 0.4 (0.0–1.2) | 8/625 | 1 (0.2–1.8) | 12/731 | 2 (0.6–3.5) | 22/1674 | 1.3 (0.6–2.0) |

On multivariable analysis, children who were orphans (adjusted odds ratio [aOR] 2.2; 95% CI: 1.0–4.8) and those with caregivers who had no knowledge of their HIV status aOR 2.4 (95% CI: 1.1–5.4) were higher odds to be HIV positive compared to those with both parents alive and those whose caregivers had been tested and knew their HIV status. Additionally, children who had been tested for HIV before the survey (aOR, 27.1; 95% CI: 11.0–66.7) and those with unknown prior testing history (aOR, 5.8; 95% CI: 2.1–16.6) were higher odds to test positive for HIV compared to those who had never been tested (Table 3).

## HIV clinical cascade

Among the HIV-positive children identified during KENPHIA 2018, 78.9% were previously known to be HIV positive (95% CI: 67.1%–90.2%). Population-level analysis also referred to the unconditional analysis of the entire population the clinical cascade was as follows; 73.6% (95% CI: 60.9%–86.2%) were receiving ART, and of these, 49.4% (95% CI: 32.1%–66.7%) were virally suppressed. However, the clinical cascade for the children with known HIV-positive or

**Table 3. Factors associated with HIV prevalence among children aged 0–14 years, KENPHIA 2018.**

| | | Child HIV status | Unadjusted odds ratios | | | Adjusted odds ratios | | |
|---|---|---|---|---|---|---|---|---|
| Characteristic | Total | Positive, n (Weighted %) | OR (95% CI) | P-value | Global p-value | OR (95% CI) | P-value | Global p-value |
| **Age category (years)** | | | | | | | | |
| 0–4 | 2457 | 11 (0.4) | ref | | | | | |
| 10–14 | 2561 | 26 (1.1) | 2.8 (1.2–6.8) | 0.02 | 0.05 | 1.7 (0.7–4.0) | 0.24 | 0.49 |
| 5–9 | 2847 | 20 (0.8) | 2.1 (0.7–6.3) | 0.18 | | 1.5 (0.5–4.9) | 0.44 | |
| **Sex** | | | | | | | | |
| Female | 3775 | 25 (0.7) | ref | | | | | |
| Male | 4090 | 32 (0.8) | 1.1 (0.5–2.6) | 0.74 | 0.74 | 1.2 (0.5–2.8) | 0.66 | 0.66 |
| **Residence** | | | | | | | | |
| Urban | 2272 | 12 (0.4) | ref | | | | | |
| Rural | 5593 | 45 (0.8) | 1.9 (0.6–5.9) | 0.26 | 0.26 | | | |
| **Orphaned/vulnerable child** | | | | | | | | |
| Both parents alive | 6802 | 37 (0.5) | ref | | | | | |
| Mother, Father, or Both dead | 846 | 20 (2.5) | 5.0 (2.5–10.3) | < .001 | < .001 | 2.2 (1.0–4.8) | 0.03 | 0.03 |
| **Child ever tested for HIV** | | | | | | | | |
| No | 5825 | 7 (0.1) | ref | | | | | |
| Don't know | 226 | 2 (1.1) | 10.8 (4.7–25.1) | < .001 | < .001 | 5.8 (2.1–16.6) | < .001 | < .001 |
| Yes | 1522 | 46 (2.8) | 28.1 (11.3–69.8) | < .001 | | 27.1 (11.0–66.7) | < .001 | |
| **Caregiver HIV status** | | | | | | | | |
| Known | 6191 | 35 (0.5) | ref | | | | | |
| Don't know | 1674 | 22 (1.3) | 2.4 (1.1–5.4) | 0.03 | 0.03 | 2.4 (1.1–5.4) | 0.02 | 0.02 |

the conditional clinical cascade was as follows: 92.0% (95% CI: 84.4%–100.0%) were receiving ART, and, of these, 67.1% (95% CI: 45.1%–89.2%) were virally suppressed (Fig 1).

## Discussion

Our study estimated 0.7% HIV prevalence among children aged 0–14 years in Kenya, translating to approximately 138,900 children living with HIV nationwide. These findings were comparable to UNAIDS-modeled 2018 estimates of pediatric HIV prevalence (0.7%) and number of CLHIV (95,000–165,000) [11], for Kenya and similar to estimates produced by other nationally-representative PHIA surveys in the region, including an estimated pediatric HIV prevalence of 0.7% in Uganda and 0.5% in Tanzania [12, 13]. Our findings also mark a

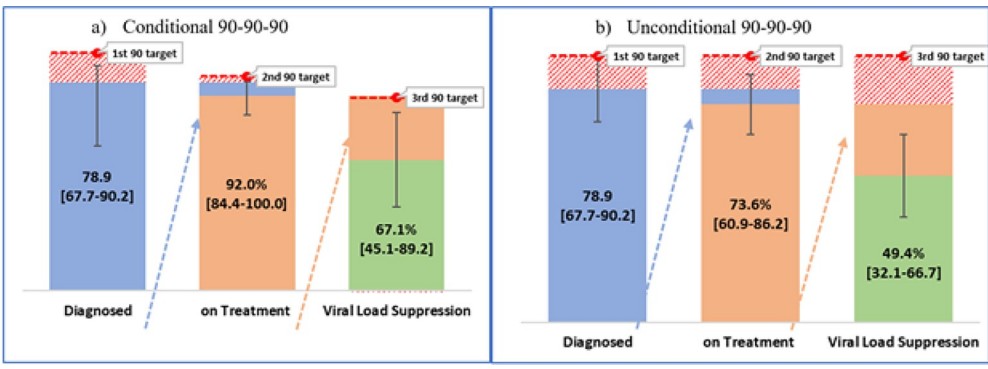

**Fig 1.** 90-90-90 HIV cascade among children living with HIV, KENPHIA 2018, a) conditional; b) unconditional.

reduction in pediatric HIV prevalence from 0.9% estimated by the KAIS 2012 [7]. Although KAIS 2012 did not include children aged <18 months [7], the observed decrease is consistent with global and regional trends in new pediatric HIV infections [14], and estimated MTCT rates in Kenya, which decreased from 14.0% in 2013 to 10.8% in 2017 and then plateaued in the last 4 years [6, 15]. These decreases in MTCT rates have largely been attributed to scale-up of PMTCT programs and improved access to ART among HIV-positive [6]. Despite the relatively low prevalence of HIV among children aged 0–14 years in Kenya, findings from our study highlight concerning persistent gaps in PMTCT and the continued need to serve the many CLHIV in pediatric HIV programs to reduce mortality and morbidity [16, 17].

We found that HIV prevalence among children in Kenya increased with age. Although not statistically significant, the higher prevalence observed among children aged 5–9 and 10–14 years in this survey is consistent with the scale-up of PMTCT and pediatric HIV programs, which has decreased MTCT and improved survival among CLHIV. Results from the survey suggest decreasing MTCT rates, consistent with the recent scale-up of universal ART for all HIV-positive pregnant and breastfeeding women.

Approximately 10% of children participating in KENPHIA 2018 were orphans, with a greater proportion of orphans among children aged 10–14 years than 0–4 years. In one study, HIV prevalence was 5 times higher among children with one or both parents deceased; after adjusting for other factors, orphans were 3 times higher odds to be HIV-positive than children with two living parents [18]. Together, these findings are consistent with the higher mortality rates observed among HIV-positive vs HIV-negative adults (parents) and improved survival among people living with HIV in recent years because of ART scale-up. HIV-related mortality in Kenya decreased by half from 2011 to 2017 following implementation of the ART initiation soon after HIV diagnosis; '*Test and Start*' A strategy in 2016 [15, 19]. Kenya has made remarkable achievements in mitigating the economic and social impact of HIV and AIDS on children and families over the past decade, but children orphaned by AIDS or who live with HIV-positive caregivers continue to face an increased risk of physical and emotional abuse and other challenges [20]. These vulnerabilities can increase the risk of HIV acquisition among HIV-negative children and the risk of poor health outcomes among CLHIV [21]. There is, therefore, need to increase the coverage of HIV programs focusing on orphans and vulnerable children to reduce vulnerability to HIV among children. Since 2017, Kenya has continued to scale up the national orphans and vulnerable children program, with approximately 599,000 children enrolled in services in 2020 [22]. To promote case management and psychosocial and economic support for children and adolescents living with HIV, Kenya has increased focus on enrolling CLHIV in appropriate services in Kenya. In 2020, 70,899 of 129,000 CLHIV in high HIV burden counties were linked to orphans and vulnerable children services in Kenya [22].

Knowledge of HIV status among HIV-positive children has doubled from 39% to 78.9% in the period between the KAIS 2012 [7], and KENPHIA 2018 surveys. This improvement coincides with aggressive scale-up of case-finding efforts in Kenya in recent years, including the Accelerated Children's Treatment (ACT) initiative established in 2015. The ACT initiative increased ART coverage for CLHIV in Kenya from 63% in 2014 to 73% in 2015, primarily through increased early infant diagnosis, pediatric HIV testing in outpatient departments and TB and malnutrition clinics and testing of children with an HIV-positive parent or sibling [23]. Despite substantial progress toward achieving the first 90 UNAIDS target in Kenya, our findings underscore the urgent need to close the remaining gap in HIV diagnosis as a key first step to ensuring all HIV-positive children have access to life-saving treatment. Thorough exploration of additional barriers to pediatric case finding in Kenya could guide targeted strategies to improve uptake of EID services.

 

We found that ART uptake among HIV-positive children with known status was 92% in our study; an increase of 20% from 2012 [7]. Although some studies in sub-Saharan Africa have shown substantial client attrition rates between the time of HIV diagnosis and initiation of ART, program data in Kenya indicates very high linkage to ART (> 90%) among HIV-positive children aged <15 years overall compared to adults [22, 24]. There are multiple factors that potentially contribute to high ART coverage among CLHIV in Kenya, including the roll-out of universal ART in 2016 and national program efforts to accelerate the identification and linkage of CLHIV to treatment [19, 23]. Although linkage to ART is high, further efforts and strategies to close the remaining gap in linkage to ART could reduce morbidity, improve health outcomes, and avoid preventable deaths among children diagnosed with HIV [25].

VL suppression was 67% among children receiving ART; our study showed some improvement since 2012 where only 50% of children on ART had attained viral suppression [7]. Significant improvements in the quality of care for CLHIV have been made in recent years, including the implementation of ART test-and-treat, establishment of age- and weight-appropriate ART dosing for children, improved disclosure rates, development of age-specific strategies for adherence support, and the use of routine VL testing for ART monitoring. Despite the increase in VL suppression among children in Kenya, the KENPHIA 2018 estimate falls well below the third 90 UNAIDS target for children receiving ART. Children who are not virally suppressed have poor health outcomes including increased risk for TB and other opportunistic infections, failure to thrive, and mortality [26, 27]. Other PHIAs showed similarly low viral suppression among children, with 45% in Uganda and 48% in South Africa [13, 28, 29].

Several care-giver, drug-related, psychosocial and patient-level barriers undermine achievement of long-term viral suppression among children receiving ART. Children depend on caregivers to support their adherence to treatment; when a caregiver has virologic failure, the risk of non-suppression in the child increases substantially [30]. To improve caregiver support of CLHIV, in 2017 Kenya adopted a family-centered model of care called Papa and Mama (PAMA) care, which provides caregiver education on supporting CLHIV, assesses and addresses specific barriers to adherence, and synchronizes caregiver and child clinic visits [31, 32]. Data from PAMA care shows high viral suppression among child -caregiver pairs at 98% and a 12 month retention in care at 99.8% [32].

Poor palatability of drugs [27, 33], and suboptimal ART regimens and inappropriate dosing among children also contribute to lower viral suppression [34]. In 2018, up to one-third of CLHIV in Kenya were receiving nevirapine-based ART, and three-quarters were receiving non-nucleoside reverse transcriptase inhibitor-based ART, both of which have low genetic barriers and, therefore, increased risk of HIV drug resistance [27, 35].

For adolescents, school schedules, delayed disclosure, and developmental changes contribute to low adherence and low viral suppression [36, 37]. Kenya has made substantial gains in improving uptake of disclosure. An assessment of adolescent HIV services in Kenya in 2018 showed high uptake (91%) of HIV disclosure among adolescents aged 13–19 years [38]. To circumvent low risk perception, peer pressure, and low self-esteem among adolescents, Kenya implemented an asset-based approach to empower adolescents and young people living with HIV to be self-health managers, to boost self-esteem, and to tap into positive peer pressure to attend scheduled appointments and adhere to ART. The initiative Operation Triple Zero has resulted to improved viral suppression among adolescents from a baseline of 63% in 2016 to 87% in 2020 [39].

Our study is subject to several limitations. First, child sampling was done in every third household, substantially decreasing the number of children sampled. In addition, 13% of children who were eligible did not provide blood samples or have valid HIV test results, posing a possible risk of selection bias. After sampling, only 57 HIV-positive children were identified.

The small number of HIV-positive children identified in the survey limits the precision of HIV prevalence estimates in both the crude and stratified analysis and limits the power to detect statistically significant differences between groups (e.g., among age groups) and among HIV risk factor variables included in the analysis. Even after stratification, the CI obtained from the pediatric analysis were too wide to provide precise estimates. Knowledge of child's HIV status was based on caregiver report and blood antiviral drug levels; as such, children with known HIV status who were not initiated on or had interruptions in ART and whose HIV status was reported as negative or unknown by the caregiver could have been misclassified as having "unknown HIV status" at the time of the survey, thereby impacting estimates of knowledge of HIV status and VL suppression among children receiving ART. To mitigate potential information bias and participant misclassification, study questionnaires were carefully designed using plain language and were conducted in the participant's preferred language by trained interviewers.

Despite these limitations, the KENPHIA 2018 survey enabled nationally representative estimates of HIV prevalence, the HIV clinical cascade, and achievement of the UNAIDS 90-90-90 targets for epidemic control among children aged 0–14 years in Kenya. Findings from the survey demonstrate the impact of Kenya's PMTCT and pediatric HIV programs since the KAIS 2012 [7], including interventions implemented to decrease MTCT, improve case finding, link CLHIV to ART, and promote VL suppression among CLHIV over the last 5 years. These findings also can help identify critical programmatic gaps for program managers, donors, and other stakeholders and can suggest areas for prioritization in the Kenya HIV program—specifically strengthening PMTCT and improving early infant diagnosis and the clinical management of children receiving ART to sustain VL suppression [40].

## Supporting information

**S1 Fig.** a. HIV testing algorithm for participants 18 months–14 years old, KENPHIA, 2018. b. HIV testing algorithm for participants <18months old, KENPHIA, 2018.
(TIF)

**S2 Fig. Patient flow chart, KENPHIA household survey, Kenya, 2018.**
(TIF)

## Acknowledgments

We acknowledge the Kenya Ministry of Health, the Ministry of Planning and Devolution, and all her partners for the scientific, strategic, and technical leadership through KENPHIA Protocol leadership team. The various planning organs of the KENPHIA through the National Executive Steering Committee the Kenya Director General of Health, the KENPHIA Secretariat, the Data Analysis and Advisory Committee and the KENPHIA Technical Working Group and Technical Sub-Committees drawn from relevant survey partner institutions. The operational support from the Council of Governors (CoG) and the 47 County Governments through the County Executive Committee Members for Health, County Directors for Health, County AIDS and STI Coordination Officers and County Medical Laboratory Coordinators. The KENPHIA study team that worked tirelessly to collect high quality data. Kenyans across the country who participated in this survey. Lastly, the unequivocal technical support of CDC and ICAP at Columbia University who made this survey possible through the financial support from PEPFAR and the Global Fund.

## Author Contributions

**Formal analysis:** Jacques Muthusi.

**Writing – original draft:** Immaculate Mutisya, Evelyn Muthoni, Raphael O. Ondondo, Jacques Muthusi, Charlotte Pahe, Lucy Nganga.

**Writing – review & editing:** Immaculate Mutisya, Evelyn Muthoni, Raphael O. Ondondo, Jacques Muthusi, Lennah Omoto, Charlotte Pahe, Abraham Katana, Evelyn Ngugi, Kenneth Masamaro, Leonard Kingwara, Trudy Dobbs, Megan Bronson, Hetal K. Patel, Nicholas Sewe, Doris Naitore, Kevin De Cock, Catherine Ngugi, Lucy Nganga.

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
