## [Decision Letter · Decision Letter 0]

2 Aug 2022

PONE-D-22-05491A national household survey on HIV prevalence and clinical cascade among children aged ≤15 years in Kenya (2018)PLOS ONE

Dear Dr. Mutisya,

Thank you for submitting your manuscript to PLOS ONE. After careful consideration, we feel that it has merit but does not fully meet PLOS ONE’s publication criteria as it currently stands. Therefore, we invite you to submit a revised version of the manuscript that addresses the points raised during the review process.

We look forward to receiving your revised manuscript.

Kind regards,

Caroline Kingori

Academic Editor

PLOS ONE

Journal Requirements:

“This survey was supported by the U.S. President’s Emergency Plan for AIDS Relief (PEPFAR) through the United States Centers for Disease Control and Prevention (CDC) under the terms of Cooperative Agreement #U2GGH001226. Additional support was provided by United States Agency for International Development (USAID) and the Global Fund to Fight AIDS, TB and Malaria.”

3. PLOS requires an ORCID iD for the corresponding author in Editorial Manager on papers submitted after December 6th, 2016. Please ensure that you have an ORCID iD and that it is validated in Editorial Manager. To do this, go to ‘Update my Information’ (in the upper left-hand corner of the main menu), and click on the Fetch/Validate link next to the ORCID field. This will take you to the ORCID site and allow you to create a new iD or authenticate a pre-existing iD in Editorial Manager. Please see the following video for instructions on linking an ORCID iD to your Editorial Manager account: " ext-link-type="uri" xlink:type="simple">https://www.youtube.com/watch?v=_xcclfuvtxQ"

Reviewers' comments:

Reviewer's Responses to Questions

**Comments to the Author**

1. Is the manuscript technically sound, and do the data support the conclusions?

Reviewer #1: Yes

Reviewer #2: Yes

2. Has the statistical analysis been performed appropriately and rigorously? 

Reviewer #1: Yes

Reviewer #2: Yes

3. Have the authors made all data underlying the findings in their manuscript fully available?

Reviewer #1: Yes

Reviewer #2: Yes

4. Is the manuscript presented in an intelligible fashion and written in standard English?

Reviewer #1: Yes

Reviewer #2: Yes

5. Review Comments to the Author

Reviewer #1: Overall well written and easy to understand. I would suggest adding a bit more demographic information earlier on to help with the flow of the paper. A lot of information is introduced in the method section e.g orphaned or living with parents, and guardian HIV status. Those can be briefly mentioned in abstract or introduction sections.

There are a few punctuation issues to be corrected as well.

Reviewer #2: I enjoyed the opportunity to review this manuscript, most notably, the strides made in tackling HIV prevalence and care among CLHIV. The use of a nationally representative sample provided great insight into who is most vulnerable and supports what is known about orphaned children. Overall a well written manuscript that outlines the need for continued efforts among those most at risk; rural and orphaned children. Insightful recommendations that those working in policy and in the community can use to identify those most vulnerable and target interventions effectively.

My feedback is really on structure and writing to make sure it is readable and findings easy to review.

• Tables 1 2 – work to make sure the titles are on one line somehow for easier reading, and maybe remove the “( )” in the 95% CI column

• Figures, I’d recommend those go/stay in the appendix section to avoid having an overly long manuscript

• Discussion section – a great job focusing on the study implications on orphaned children and the various initiatives implemented to tackle these, but maybe add a line on how effective these programs have been (given that your study shows this demographic continues to be at a significantly higher risk) and a recommendation for improvement based on your studies. You do this well in your next paragraph focused on adolescents to showcase effectiveness of the Operation Triple Zero.

• Writing – extra word “the” noted in data analysis paragraph; 3rd line from the bottom “The protocol was reviewed and cleared the by the Institutional”. As always be sure to review entire document to catch those hidden errors.

6. PLOS authors have the option to publish the peer review history of their article (what does this mean?). If published, this will include your full peer review and any attached files.

Reviewer #1: No

Reviewer #2: No

---

## [Author Response · Author response to Decision Letter 0]

16 Sep 2022

Thank you for reviewing and giving us the recommendations to our manuscript 

Responses to the academic editor 

• We have provided a funding statement which includes all funding sources as follows:-

“This survey was supported by the U.S. President’s Emergency Plan for AIDS Relief (PEPFAR) through the United States Centers for Disease Control and Prevention (CDC) under the terms of Cooperative Agreement #U2GGH001226. Additional support was provided by United States Agency for International Development (USAID) and the Global Fund to Fight AIDS, TB and Malaria.” There was no additional external funding received for this study.

• Add ORCID ID the corresponding author 

We have included the ORCID Id for the corresponding author

• Review reference list 

We have reviewed our reference list and ensured that it is complete and correct. We have not referenced any retracted papers

Responses to the peer reviewers

Reviewer #1: 

• Overall well written and easy to understand. I would suggest adding a bit more demographic information earlier on to help with the flow of the paper. A lot of information is introduced in the method section e.g orphaned or living with parents, and guardian HIV status. Those can be briefly mentioned in abstract or introduction sections. There are a few punctuation issues to be corrected as well. Thank you for this comment. We have corrected the punctuation errors and included information on the burden of AIDS orphans in the introduction . ’

• In Kenya, an estimated 690,000 children are orphaned by AIDS [5]. AIDS orphans who are HIV infected, live under the care of relatives or children homes.

Reviewer #2: 

• I enjoyed the opportunity to review this manuscript, most notably, the strides made in tackling HIV prevalence and care among CLHIV. The use of a nationally representative sample provided great insight into who is most vulnerable and supports what is known about orphaned children. Overall, a well written manuscript that outlines the need for continued efforts among those most at risk; rural and orphaned children. Insightful recommendations that those working in policy and in the community can use to identify those most vulnerable and target interventions effectively. 

• My feedback is really on structure and writing to make sure it is readable and findings easy to review. Tables 1 2 – work to make sure the titles are on one line somehow for easier reading, and maybe remove the “( )” in the 95% CI column 

• Thank you very much for picking this. We have corrected the tables.

• Figures, I’d recommend those go/stay in the appendix section to avoid having an overly long manuscript

• Figure 1 and Figure 2 renamed as Supplementary figures (to come at the end of the manuscript). Renamed Figure 3 as Figure 1 and retained it to appear in the main paper 

• Discussion section – a great job focusing on the study implications on orphaned children and the various initiatives implemented to tackle these, but maybe add a line on how effective these programs have been (given that your study shows this demographic continues to be at a significantly higher risk) and a recommendation for improvement based on your studies. You do this well in your next paragraph focused on adolescents to showcase effectiveness of the Operation Triple Zero. 

• Thank you for the comment. We have included information on PAMA care success . ‘Data from PAMA care shows high viral suppression among child -caregiver pairs at 98% and a 12 month retention in care at 99.8% [32]’.

• Writing – extra word “the” noted in data analysis paragraph; 3rd line from the bottom “The protocol was reviewed and cleared the by the Institutional”. As always be sure to review entire document to catch those hidden errors. 

Thank you this is corrected.

---

## [Decision Letter · Decision Letter 1]

1 Nov 2022

A national household survey on HIV prevalence and clinical cascade among children aged ≤15 years in Kenya (2018)

PONE-D-22-05491R1

Dear Dr. Mutisya,

We’re pleased to inform you that your manuscript has been judged scientifically suitable for publication and will be formally accepted for publication once it meets all outstanding technical requirements.

Kind regards,

Caroline Kingori

Academic Editor

PLOS ONE

Additional Editor Comments (optional):

Authors provided great insight on the prevailing pediatric HIV incidence and prevalence rates in Kenya. It is concerning to see the staggering increase in pediatric HIV infections despite the success that Kenya has had in curtailing incidence rates across the populace. I am curious about the prevailing challenges post COVID-19 and how that has affected pediatric HIV infections. That could be a follow-up paper. For now, I support the reviewers comment and recommend the paper be accepted for publication.

Reviewers' comments:

Reviewer's Responses to Questions

**Comments to the Author**

1. If the authors have adequately addressed your comments raised in a previous round of review and you feel that this manuscript is now acceptable for publication, you may indicate that here to bypass the “Comments to the Author” section, enter your conflict of interest statement in the “Confidential to Editor” section, and submit your "Accept" recommendation.

Reviewer #1: All comments have been addressed

Reviewer #2: (No Response)

2. Is the manuscript technically sound, and do the data support the conclusions?

Reviewer #1: Yes

Reviewer #2: (No Response)

3. Has the statistical analysis been performed appropriately and rigorously? 

Reviewer #1: Yes

Reviewer #2: (No Response)

4. Have the authors made all data underlying the findings in their manuscript fully available?

Reviewer #1: Yes

Reviewer #2: (No Response)

5. Is the manuscript presented in an intelligible fashion and written in standard English?

Reviewer #1: Yes

Reviewer #2: (No Response)

6. Review Comments to the Author

Reviewer #1: Succinct and thorough research. The writers addressed all review concerns and I have recommend the paper for publishing.

Reviewer #2: (No Response)

7. PLOS authors have the option to publish the peer review history of their article (what does this mean?). If published, this will include your full peer review and any attached files.

Reviewer #1: No

Reviewer #2: No

---

## [Editor Report · Acceptance letter]

14 Nov 2022

PONE-D-22-05491R1 

A national household survey on HIV prevalence and clinical cascade among children aged ≤15 years in Kenya (2018) 

Dear Dr. Mutisya:

I'm pleased to inform you that your manuscript has been deemed suitable for publication in PLOS ONE. Congratulations! Your manuscript is now with our production department. 

Kind regards, 

on behalf of

Dr. Caroline Kingori 

Academic Editor

PLOS ONE